# Different patterns of white matter microstructural alterations between psychotic and non-psychotic bipolar disorder

**Dong-Kyun Lee[1], Hyeongrae Lee[1], Vin Ryu[1], Sung-Wan Kim[2], Seunghyong Ryu[2]***

1 Department of Mental Health Research, National Center for Mental Health, Seoul, Republic of Korea,
2 Department of Psychiatry, Chonnam National University Medical School, Gwangju, Republic of Korea

* ryush@chonnam.ac.kr

**Data Availability Statement:** The preprocessed DTI data and de-identified clinical data were deposited in the Dryad Digital Repository: https://datadryad.org/stash/share/

## Abstract

This study aimed to investigate alterations in white matter (WM) microstructure in patients with psychotic and non-psychotic bipolar disorder (PBD and NPBD, respectively). We used 3T-magnetic resonance imaging to examine 29 PBD, 23 NPBD, and 65 healthy control (HC) subjects. Using tract-based spatial statistics for diffusion tensor imaging data, we compared fractional anisotropy (FA) and mean diffusion (MD) pairwise among the PBD, NPBD, and HC groups. We found several WM areas of decreased FA or increased MD in the PBD and NPBD groups compared to HC. PBD showed widespread FA decreases in the corpus callosum as well as the bilateral internal capsule and fornix. However, NPBD showed local FA decreases in a part of the corpus callosum body as well as in limited regions within the left cerebral hemisphere, including the anterior and posterior corona radiata and the cingulum. In addition, both PBD and NPBD shared widespread MD increases across the posterior corona radiata, cingulum, and sagittal stratum. These findings suggest that widespread WM microstructural alterations might be a common neuroanatomical characteristic of bipolar disorder, regardless of being psychotic or non-psychotic. Particularly, PBD might involve extensive inter-and intra-hemispheric WM connectivity disruptions.

## 1. Introduction

Bipolar disorder (BD) is a disabling neuropsychiatric illness characterized by recurrent manic and depressive episodes [1]. Since Kraepelin's concept of manic-depressive insanity, BD had historically been considered a mood disorder along with major depressive disorder [2]. However, BD is highly heterogeneous in its clinical manifestations, often presenting schizophrenia-like psychotic symptoms as well as mood dysregulation [3]. Moreover, large-scale genetic studies have provided strong evidence for common genetic causes between BD and schizophrenia [4, 5], while neuroimaging studies have shown shared structural abnormalities between the two disorders [6]. Considering this evidence, the concept of a psychosis continuum linking BD and schizophrenia has been recently highlighted [7, 8].

b77jfKwD4up4Zd3UM1L9ePFEEoR2-J-NU1rwA7XWHaA.

**Funding:** This research was supported by an intramural research grant from the Republic of Korea's National Center for Mental Health (R2020-B). The funders had no role in study design, data collection and analysis, decision to publish, or preparation of the manuscript.

**Competing interests:** The authors have declared that no competing interests exist.

A significant proportion of patients (approximately 70% of patients with BD type I) have experienced frank psychotic symptoms, such as hallucination and delusion, during their illness [9]. Particularly, compared to non-psychotic BD (NPBD), psychotic BD (PBD) is characterized by earlier onset, poorer treatment response, more relapse, and lower cognitive and social function, also common in patients with schizophrenia [10]. Thus, from the point of view of a psychosis continuum, PBD has been expected to be closer to psychotic disorders such as schizophrenia, whereas NPBD might be closer to affective disorders [11]. However, the degree of shared biological underpinnings between PBD and schizophrenia and the differences in neuroanatomical characteristics between PBD and NPBD remain unclear. Accordingly, exploring the neural substrates of the psychotic features in BD could improve our understanding of the neurobiological basis of the psychosis continuum clinical concept.

Diffusion tensor imaging (DTI) studies have revealed extensive white matter (WM) abnormalities not only in schizophrenia [12] but also in BD [13, 14]. Recent comparative studies evidenced an overlap of affected WM areas in BD and schizophrenia [15, 16]. However, few studies have investigated the neural substrates associated with psychotic features in BD, and little is known about the patterns of WM abnormalities in PBD and NPBD. Therefore, this study aimed to concurrently investigate WM alterations in PBD and NPBD using tract-based spatial statistical (TBSS) analyses and explore WM microstructural differences between the two BD subtypes. Particularly, we hypothesized that PBD might involve extensive WM abnormalities, possibly comparable to those of schizophrenia. This is in line with evidence of more extensive gray matter volume deficits in PBD than in NPBD [17, 18].

## 2. Materials and methods

### 2.1. Subjects

All subjects were recruited from outpatient clinics of the National Center for Mental Health, Seoul, Korea, and a psychologist interviewed them using the Mini-International Neuropsychiatric Interview (MINI) [19]. The diagnosis of BD was assessed through the Diagnostic and Statistical Manual of Mental Disorders, Fourth Edition (DSM-IV) criteria and confirmed by the clinical consensus of expert psychiatrists. Inclusion criteria were as follows: (1) age 20–50 years old; (2) duration of illness > 1 year; (3) no change in general clinical state and medication for ≥3 months prior to the time of assessment. We excluded patients with a concurrent axis I diagnosis according to the DSM-IV including schizophrenia, schizoaffective disorder, and alcohol use disorder, current or past neurological disease, any contraindication to MRI scan, or a physical condition that would render an MRI scan difficult to administer or interpret. According to the K module in the MINI [19], patients with lifetime experience of psychotic symptoms were assigned into the PBD group, while the others were assigned to the NPBD group. The healthy control (HC) group consisted of volunteers from the local community in the same age range and without history of psychiatric disorders. The same exclusion criteria for the patients was also applied to the HC group with regards to medical, neurological, and physical conditions. We assessed the patients' overall psychopathology and functioning level using the 18-item Brief Psychiatric Rating Scale (BPRS-18) [20], Clinician-Rated Dimensions of Psychosis Symptom Severity (CRDPSS) [21], and WHO Disability Assessment Schedule 2.0 (WHODAS 2.0) [22].

We first selected 31 PBD, 31 NPBD, and 65 HC, who were recruited in order, except for three patients with BD who were excluded due to poor information or imaging. According to the MINI, among the patients, two PBD and eight NPBD were classified as BD type II, and the remaining were classified as BD type I. In addition, none of the patients had taken any

substance that could induce psychosis. For subject homogeneity, we included only 52 patients with BD type I. Finally, 29 PBD, 23 NPBD, and 65 HC were included in this study.

This study was approved by the Institutional Review Board of the National Center for Mental Health (IRB approval number: 116271-2017-26), and written informed consent obtained from all subjects.

## 2.2. MRI data acquisition

MRI data were acquired using a 3-Tesla MRI scanner (Ingenia CX; Philips, Erlangen) equipped with a 32-channel head coil at National Center for Mental Health. A diffusion-weighted image was acquired using single-shot echo-planar imaging sequence with the following parameters: acquisition matrix = $128 \times 128$, voxel size = $1.75 \times 1.75 \times 2$ mm$^3$, axial slices = 72, FOV = $224 \times 224$ mm$^2$, TE = 90 ms, TR = 9000 ms, flip angle = 90˚, slice gap = 0 mm, b-value = 0 and 1000 s/mm$^2$, and diffusion sensitive gradient direction = 64. The baseline image without weighting was used [0, 0, 0].

## 2.3. DTI data processing

DTI processing was performed using the FMRIB Software Library (FSL) (https://fsl.fmrib.ox.ac.uk/fsl/fslwiki), ver. 6.0. First, motion artifacts and eddy current distortions through affine registration were corrected by considering the B0 volume as a reference using FSL's Diffusion Toolbox. Then, the diffusion-weighted images were skull stripped using the Brain Extraction Tool within the FSL. Fractional anisotropy (FA) and mean diffusivity (MD) images were obtained from the tensors' eigenvalues using the DTIFIT program in the FSL. Next, voxel-wise statistical analysis of FA and MD images was performed using the TBSS pipeline [23]. FA is the most widely used scalar in DTI, a measure indicating the overall directionality of water diffusion, higher in organized WM tracts and lower in disorganized fibers [24]. Axon damage or demyelination results in water motion being more isotropic, which may manifest in low FA values. MD describes the rotationally invariant magnitude of water diffusion within brain tissue, independent of tissue directionality [25]. MD is a non-specific, albeit sensitive, measure that can be affected by any disease process affecting the barriers restricting water movement and is usually higher in damaged tissue with edema or necrosis, for instance. FA images were aligned into the standard space (FMRIB58_FA, $1 \times 1 \times 1$ mm MNI 152 space) using the non-linear registration tool (FNIRT). Afterwards, a mean FA image was created and threshold by an FA value of 0.2 to exclude peripheral tracts and GM regions. Each subject's aligned FA images were then projected onto the skeleton by filling it with the highest FA values from the nearest relevant center of the fiber tracts. The same transformation and warped-field were applied to MD images.

## 2.4. Statistical analyses

TBSS analyses were performed using the FSL toolbox (randomize). Pairwise comparisons of FA and MD values were performed at the voxel-level for PBD vs. NPBD, PBD vs. HC, NPBD vs. HC, and BD (PBD + NPBD) vs. HC using a generalized linear model. We adjusted for age, sex, education, and handedness as covariates to compare DTI parameters. To adjust the voxel-wise multiple comparisons, we adopted the family-wise error (FWE) approach. Significance thresholding for the TBSS analysis was determined using 10,000 permutations and threshold-free cluster-enhancement with the 2D parameter settings [26]. Thereafter, to adjust the number of comparisons of two DTI parameters (FA and MD) between four group pairs (PBD vs. NPBD, PBD vs. HC, NPBD vs. HC, and BD vs. HC), we applied the Bonferroni correction to the FWE-corrected P-value. Thus, the threshold of statistical significance for the TBSS

analysis was conservatively set at an FWE-corrected P-value < 0.0062 (0.05 / 8) with a cluster size > 100 mm$^3$.

In addition to group comparisons, we conducted exploratory analyses to investigate the association of DTI parameters with severity of psychopathology and functioning level, represented by the BPRS-18 and WHODAS 2.0 total scores, respectively, in each PBD, NPBD, and BD group. For these analyses, we adjusted for sex, age, education, handedness, and chlorpromazine-equivalent dose of antipsychotic drug, applying an FWE-corrected P-value < 0.05.

For comparisons of demographic and clinical data, a P-value < 0.05 was considered statistically significant.

## 3. Results

### 3.1. Demographic and clinical characteristics

Table 1 summarizes the demographic and clinical characteristics of the PBD, NPBD, and HC groups. There were no significant differences in age (F = 0.34, P = 0.711), sex ($\chi^2$ = 4.36, P = 0.113), level of education (F = 2.55, P = 0.083), and handedness ($\chi^2$ = 0.25, P = 0.884) among the three groups, as well as in the duration of illness (t = 1.05, P = 0.298) and chlorpromazine-equivalent dose of antipsychotic drugs (t = 1.70, P = 0.096) between the PBD and NPBD groups. According to the BPRS-18 and WHODAS 2.0 scores, PBD and NPBD were clinically stable without overall vivid psychotic symptoms and their social functioning was also well preserved at the time of assessments. In particular, the scores of CRDPSS domains showed that most PBD and NPBD had been experiencing less than a mild level of manic or depressive symptoms. There were also no significant differences in severity of psychopathology and functioning level between the PBD and NPBD groups. In addition, there were no significant differences in age (t = 0.83, P = 0.409), sex ($\chi^2$ = 3.94, P = 0.063), and handedness ($\chi^2$ = 0.21, P = 0.771) between the BD group and HC. However, the level of education was significantly lower in the BD group than in HC (t = -2.18, P = 0.032).

### 3.2. FA and MD comparison between groups

Comparisons of FA between PBD and NPBD revealed no significant differences. However, compared with HC, PBD showed widespread FA decreases in the body and splenium of the corpus callosum as well as in the bilateral internal capsule and fornix (Fig 1 and Table 2). NPBD showed local FA decreases in a part of the corpus callosum body and limited regions within the left cerebral hemisphere, including the anterior and posterior corona radiata and cingulum.

Compared with HC, all patients with BD also showed widespread FA decreases in the right posterior thalamic radiation, right posterior corona radiata, and left sagittal stratum as well as the corpus callosum, left retrolenticular part of the internal capsule, and the left posterior corona radiata, which were also affected in PBD and NPBD (Fig 1 and Table 2).

Comparisons of MD between PBD and NPBD revealed no significant differences. However, compared with HC, PBD and NPBD showed a widespread MD increase mainly in the right superior corona radiate (Fig 2 and Table 3). In addition, PBD showed MD increases across the left posterior corona radiata, left posterior thalamic radiation, left cingulum, left crus of fornix, and bilateral sagittal stratum. NPBD also showed MD increases across the right retrolenticular part of internal capsule, left superior longitudinal fasciculus, bilateral cingulum, and left sagittal stratum.

Compared with HC, all patients with BD also showed widespread MD increases in the column and body of the fornix and right posterior thalamic radiation as well as in the right

**Table 1. Demographic and clinical characteristics.**

| Variables[a] | Psychotic bipolar disorder (N = 29) | Non-psychotic bipolar disorder (N = 23) | Healthy control (N = 65) | Statistics[b] |
|---|---|---|---|---|
| Age, y | 35.90 ± 7.33 | 35.78 ± 8.93 | 34.52 ± 8.93 | F = 0.34, P = 0.711 |
| Sex (male / female), n | 19 / 10 | 13 / 10 | 28 / 37 | $\chi^2$ = 4.36, P = 0.113 |
| Education, y | 14.14 ± 1.77 | 13.61 ± 1.88 | 14.78 ± 2.55 | F = 2.55, P = 0.083 |
| Handedness (right / left), n | 26 / 3 | 21 / 2 | 57 / 8 | $\chi^2$ = 0.25, P = 0.884 |
| Duration of illness, y | 13.71 ± 8.15 | 11.65 ± 7.31 | - | t = 1.05, P = 0.298 |
| Antipsychotics, n | 29 | 22 | - | $\chi^2$ = 1.29, P = 0.442 |
| Chlorpromazine-equivalent dose, mg | 516.25 ± 315.08 | 370.23 ± 301.52 | - | t = 1.70, P = 0.096 |
| Mood stabilizers, n | 20 | 20 | - | $\chi^2$ = 2.34, P = 0.188 |
| valproate / lithium / lamotrigine, n | 14 / 7 / 3 | 14 / 7 / 1 | - | - |
| Antidepressants, n | 1 | 4 | - | $\chi^2$ = 2.87, P = 0.157 |
| BPRS-18 total score | 27.24 ± 7.49 | 27.74 ± 8.17 | - | t = -0.23, P = 0.820 |
| BPRS-18 subscale scores[c] | | | | |
| Affect | 7.76 ± 3.25 | 8.09 ± 3.94 | - | t = -0.33, P = 0.743 |
| Positive symptoms | 5.72 ± 3.07 | 5.48 ± 2.04 | - | t = 0.33, P = 0.743 |
| Negative symptoms | 5.21 ± 2.16 | 4.78 ± 2.43 | - | t = 0.66, P = 0.509 |
| Resistance | 3.66 ± 1.26 | 4.43 ± 1.93 | - | t = -1.68, P = 0.102 |
| Activation | 3.90 ± 1.40 | 4.09 ± 1.50 | - | t = -0.47, P = 0.639 |
| CRDPSS total score | 3.66 ± 3.24 | 4.35 ± 3.76 | - | t = -0.71, P = 0.479 |
| CRDPSS domain scores | | | | |
| Hallucination | 0 (0–0) | 0 (0–0) | - | U = 324.50, P = 0.682 |
| Delusion | 0 (0–0) | 0 (0–1) | - | U = 292.00, P = 0.313 |
| Disorganized speech | 0 (0–0) | 0 (0–1) | - | U = 306.00, P = 0.477 |
| Abnormal psychomotor behavior | 0 (0–0) | 0 (0–0) | - | U = 316.00, P = 0.425 |
| Negative symptom | 1 (0–2) | 0 (0–1) | - | U = 305.00, P = 0.566 |
| Impaired cognition | 1 (0–1) | 1 (0–2) | - | U = 261.50, P = 0.155 |
| Depression | 1 (0–2) | 1 (0–2) | - | U = 306.50, P = 0.593 |
| Mania | 0 (0–1) | 0 (0–1) | - | U = 325.00, P = 0.850 |
| WHODAS 2.0 total score | 10.01 ± 7.67 | 11.61 ± 7.45 | - | t = -0.75, P = 0.454 |
| WHODAS 2.0 domain scores | | | | |
| Cognition | 10.78 ± 11.44 | 12.86 ± 10.65 | - | t = -0.68, P = 0.500 |
| Mobility | 1.21 ± 3.18 | 3.04 ± 5.98 | - | t = -1.42, P = 0.161 |
| Self-care | 2.59 ± 3.92 | 2.72 ± 4.55 | - | t = -0.11, P = 0.912 |
| Getting along | 21.38 ± 17.57 | 21.96 ± 16.22 | - | t = -0.12, P = 0.904 |
| Life activities | 6.47 ± 7.33 | 9.65 ± 8.53 | - | t = -1.45, P = 0.154 |
| Participation | 17.67 ± 12.81 | 19.43 ± 10.23 | - | t = -0.54, P = 0.595 |

[a] Data are shown as mean ± standard deviation, number, or median (interquartile range).

[b] ANOVA, Fisher exact test, independent t test, or Mann-Whitney test.

[c] Factor structures proposed by Shafer (2005) [27].

Abbreviations: BPRS-18, 18-item Brief Psychiatric Rating Scale; CRDPSS, Clinician-Rated Dimensions of Psychosis Symptom Severity; WHODAS 2.0, WHO Disability Assessment Schedule 2.0.

superior corona radiata and left cingulum, which were also affected in PBD and NPBD (Fig 2 and Table 3).

In addition, we found no significant association between DTI parameters and continuous clinical variables, including the BPRS-18 and WHODAS 2.0 total scores, in all patients with BD as well as in PBD and NPBD.

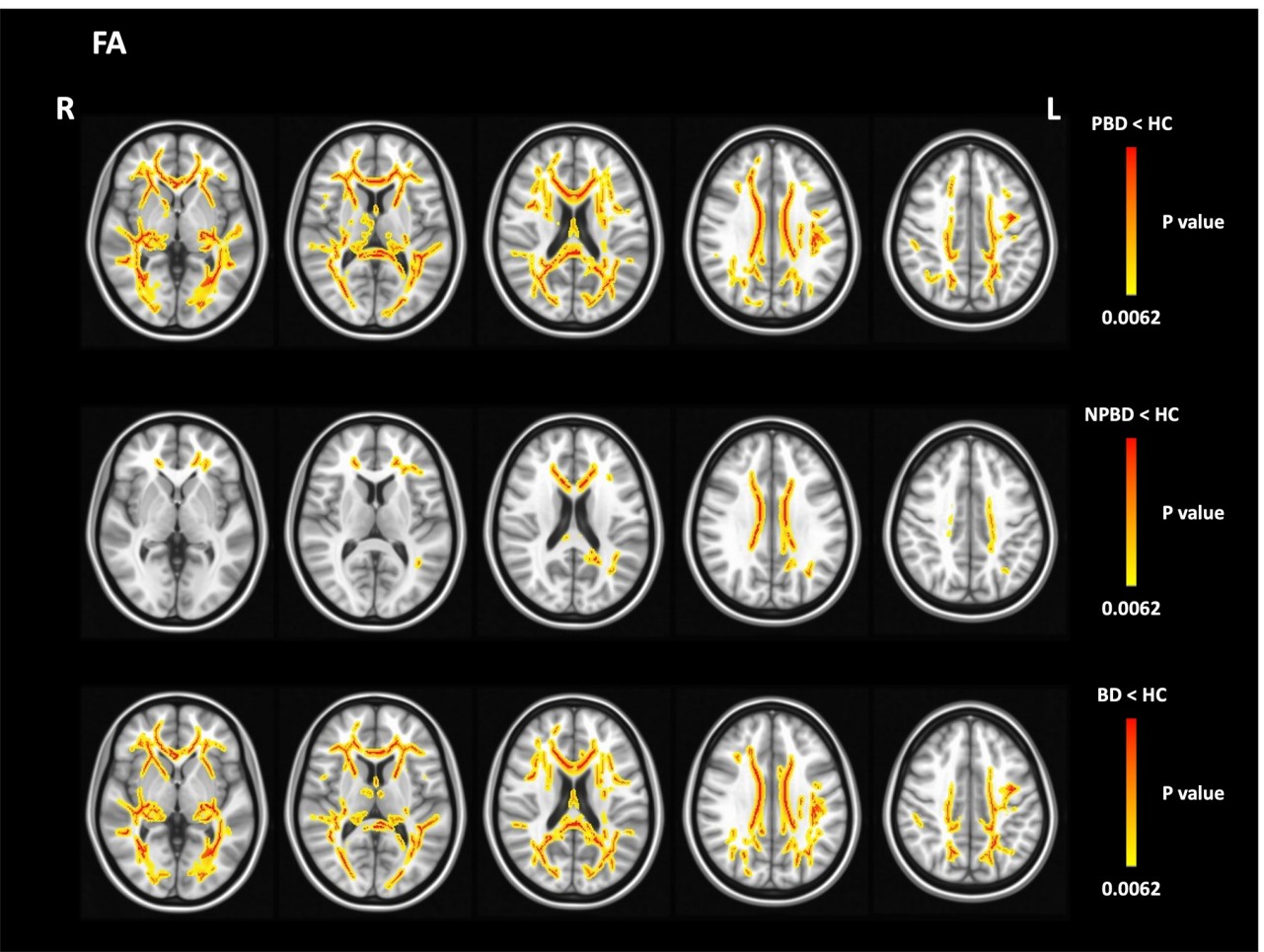

**Fig 1. Tract-based spatial statistics analyses identified white matter areas affected by a significant decrease in fractional anisotropy (FA) in patients with psychotic bipolar disorder (PBD) (N = 29), non-psychotic bipolar disorder (NPBD) (N = 23), and bipolar disorder (BD) (N = 52) in comparisons with healthy controls (HCs) (N = 65).** Family-wise error corrected P < 0.0062.

## 4. Discussion

This study investigated WM microstructural alterations in PBD and NPBD using TBSS analyses. We observed no significant difference in diffusion measures between PBD and NPBD. However, in comparison with HC, patients with PBD and NPBD exhibited widespread alterations in the WM microstructure, with possible differences between the two BD subtypes. PBD showed widespread decreases in FA, particularly in the body and splenium of the corpus callosum, as well as in the bilateral internal capsule and fornix. Conversely, NPBD showed local decreases in FA only in a part of the corpus callosum body and some WM areas within the left cerebral hemisphere. In contrast to decreased FA, both PBD and NPBD showed widespread increases in MD across the whole-brain WM skeleton.

### 4.1. Comparisons of diffusion measures between PBD and NPBD

In this study, we first compared diffusional measures between PBD and NPBD to explore WM substrates that might be unique to PBD contributing to psychotic features and poorer clinical course. However, there was no significant difference in FA and MD between PBD and NPBD.

**Table 2. Comparisons of fractional anisotropy among PBD (N = 29), NPBD (N = 23), BD (PBD + NPBD) (N = 52), and HC (N = 65) groups[a].**

| Anatomical region | Side | T max | Peak coordinates (MNI) | | | Cluster size |
|---|---|---|---|---|---|---|
| | | | x | y | z | (mm³) |
| PBD vs. HC | | | | | | |
| Body of corpus callosum | | 6.873 | -10 | 17 | 22 | 30,764 |
| Retrolenticular part of internal capsule | L | 4.707 | -29 | -24 | 5 | 1787 |
| Fornix (column and body) | | 4.300 | 0 | 6 | 6 | 625 |
| Splenium of corpus callosum | | 3.960 | -2 | -34 | 14 | 355 |
| Anterior limb of internal capsule | R | 2.728 | 12 | 5 | 4 | 120 |
| NPBD vs. HC | | | | | | |
| Body of corpus callosum | | 5.447 | -10 | 19 | 21 | 4291 |
| Posterior corona radiata | L | 5.532 | -17 | -53 | 31 | 579 |
| Anterior corona radiata | L | 4.047 | -26 | 33 | 6 | 456 |
| Cingulum | L | 5.085 | -7 | -10 | 35 | 346 |
| BD vs. HC | | | | | | |
| Body of corpus callosum | | 6.807 | -9 | 17 | 22 | 25,656 |
| Posterior thalamic radiation | R | 4.880 | 32 | -64 | 1 | 3908 |
| Retrolenticular part of internal capsule | L | 4.441 | -29 | -22 | 3 | 1451 |
| Posterior corona radiata | R | 3.598 | 37 | -57 | 23 | 423 |
| Splenium of corpus callosum | | 3.759 | -7 | -39 | 15 | 365 |
| Sagittal stratum | L | 3.813 | -49 | -24 | -17 | 286 |
| Superior longitudinal fasciculus | R | 3.030 | 42 | -43 | 9 | 165 |
| Posterior corona radiata | L | 3.618 | -26 | -33 | 25 | 115 |

[a] Family-wise error corrected P < 0.0062

Note that all MNI coordinates of maximum t values are selected in the significant region.

Abbreviations: PBD, Psychotic Bipolar Disorder; NPBD, Non-Psychotic Bipolar Disorder; BD, Bipolar Disorder; HC, Healthy Control; MNI, Montreal Neurological Institute; L, Left; R, Right.

Similar to our finding, recent studies have shown minimal differences in DTI parameters between PBD and NPBD. Ji et al. found a significant FA difference between the two BD subtypes only in the left uncinate fasciculus [28]. Furthermore, Brown et al. could not find any difference in diffusion measures between PBD and NPBD [29]. The sample size of these studies, and our study, might have been too small to detect subtle differences between the two BD subtypes. In addition, these studies revealed significant diffusional alterations in patients with PBD and NPBD compared with HC [28, 29]. These microstructural alterations have mainly been interpreted as the WM pathology associated with affective disturbance in BD. However, we supposed that the distribution pattern of WM abnormalities might be different between PBD and NPBD, which might be a neuroanatomical characteristic underlying the different clinical manifestations between the two BD subtypes.

## 4.2. Widespread compromise of WM integrity and connectivity in PBD

In the present study, compared with HC, PBD showed a pronounced FA decrease in the corpus callosum, with widespread FA decreases across the internal capsule and fornix. PBD also showed widespread WM areas affected by MD increase across the right superior corona radiata, left posterior thalamic radiation, and left cingulum. Considering that either a decrease in FA or an increase in MD reflects disruptions to water diffusion coherence due to axonal degradation, demyelination, or neurodegeneration [25], these findings suggest that PBD might involve widespread alterations in the WM microstructure across cortico-cortical, cortico-

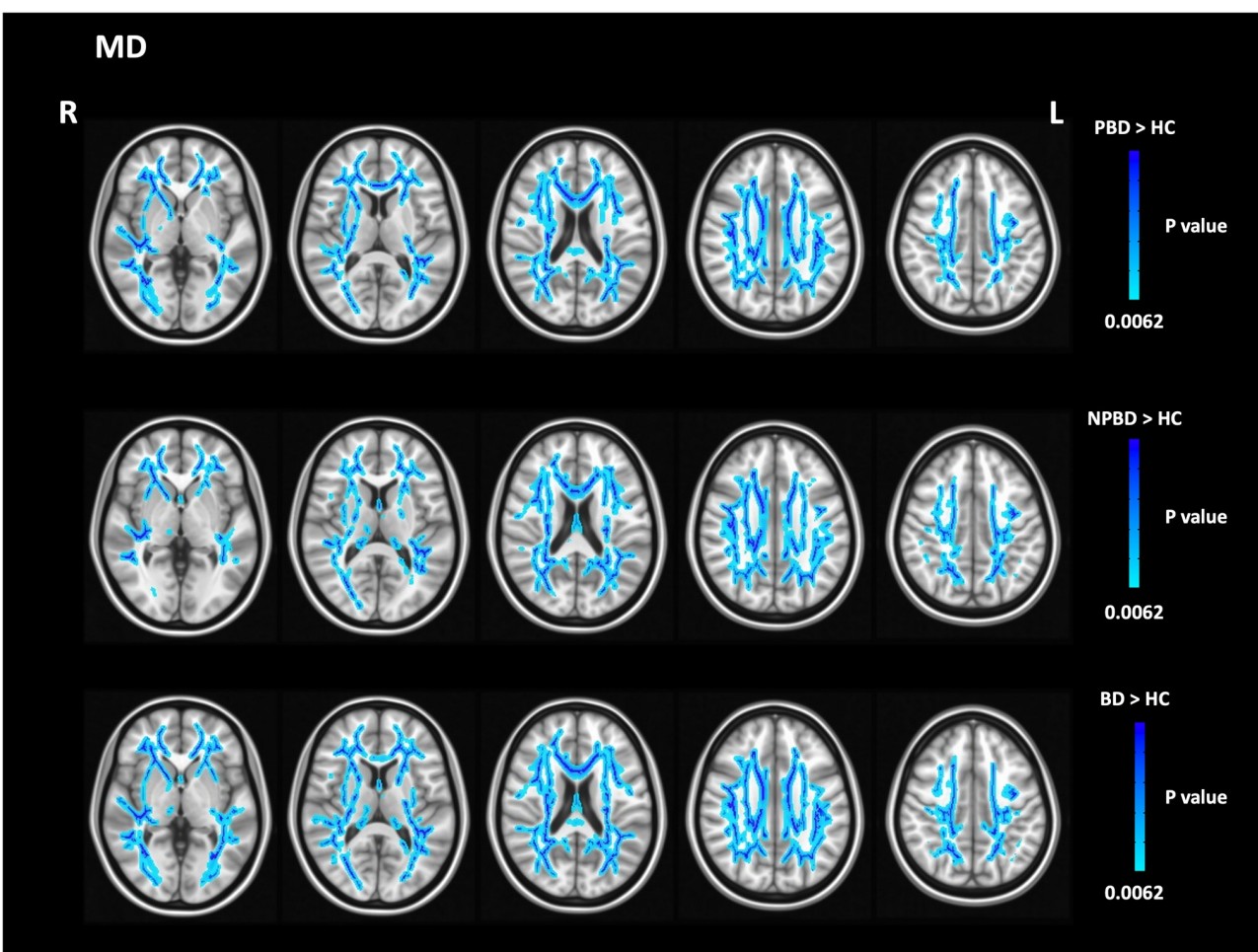

**Fig 2. Tract-based spatial statistics analyses identified white matter areas affected by a significant increase in mean diffusivity (MD) in patients with psychotic bipolar disorder (PBD) (N = 29), non-psychotic bipolar disorder (NPBD) (N = 23), and bipolar disorder (BD) (N = 52) in comparisons with healthy controls (HCs) (N = 65).** Family-wise error corrected P < 0.0062.

limbic, thalamocortical, and callosal connections. These patterns of widespread WM abnormalities are also largely in agreement with the results of previously published large DTI analyses and meta-analyses of schizophrenia, which provided evidence of extensive disruptions of WM integrity and connectivity across frontotemporal, frontosubcortical, and callosal networks in patients of schizophrenia [30–32]. In particular, given that the corpus callosum contains axon fibers connecting the bilateral frontal cortices and that WM integrity is related to cognitive performance in various domains including sustained attention, processing speed, and problem solving abilities [33], extensive disruptions of WM connectivity in the corpus callosum might be the underlying mechanism of the more severe cognitive decline in PBD compared to that in NPBD [10]. Therefore, this study's findings suggest that extensive WM integrity and connectivity disruptions might be the neural substrates underlying the clinical manifestations of PBD, such as concurrent psychotic features, cognitive impairment, and poor clinical outcome, overlapping with those of patients with schizophrenia, largely in line with our prior hypothesis that PBD might be close to schizophrenia on a psychosis continuum.

**Table 3. Comparisons of mean diffusivity among PBD (N = 29), NPBD (N = 23), BD (PBD + NPBD) (N = 52), and HC (N = 65) groups[a].**

| Anatomical region | Side | T max | Peak coordinates (MNI) | | | Cluster size |
|---|---|---|---|---|---|---|
| | | | x | y | Z | (mm³) |
| PBD vs. HC | | | | | | |
| Superior corona radiata | R | 6.704 | 28 | 7 | 27 | 35,366 |
| Posterior corona radiata | L | 5.010 | -17 | -51 | 32 | 604 |
| Posterior thalamic radiation | L | 3.911 | -36 | -52 | 11 | 429 |
| Cingulum | L | 5.045 | -8 | -5 | 33 | 416 |
| Fornix (crus) | L | 5.079 | -34 | -9 | -16 | 366 |
| Sagittal stratum | R | 4.019 | 41 | -38 | -10 | 198 |
| Sagittal stratum | L | 3.373 | -40 | -42 | -7 | 114 |
| Anterior limb of internal capsule | L | 3.649 | -22 | -4 | 16 | 107 |
| Retrolenticular part of internal capsule | L | 4.019 | -29 | -23 | -1 | 103 |
| NPBD vs. HC | | | | | | |
| Superior corona radiata | R | 6.222 | 28 | 6 | 29 | 27,510 |
| Retrolenticular part of internal capsule | R | 5.217 | 41 | -29 | 3 | 912 |
| Superior longitudinal fasciculus | L | 5.466 | -44 | -1 | 27 | 748 |
| Cingulum | L | 4.705 | -11 | -48 | 23 | 377 |
| Sagittal stratum | L | 3.723 | -40 | -29 | -16 | 270 |
| External capsule | R | 4.480 | 32 | -2 | 12 | 169 |
| Cingulum | R | 4.436 | 9 | 16 | 28 | 141 |
| Fornix (crus) | R | 5.250 | 11 | -47 | 25 | 107 |
| BD vs. HC | | | | | | |
| Superior corona radiata | R | 7.465 | 28 | 7 | 28 | 40,579 |
| Fornix (column and body) | | 5.579 | 3 | -11 | 15 | 783 |
| Cingulum | L | 4.893 | -8 | -5 | 33 | 489 |
| Posterior thalamic radiation | R | 4.394 | 36 | -56 | -6 | 339 |
| Retrolenticular part of internal capsule | L | 3.189 | -29 | -23 | -1 | 186 |
| External capsule | R | 4.602 | 35 | -5 | -1 | 143 |
| External capsule | L | 3.453 | -31 | -16 | 12 | 129 |
| Superior corona radiata | L | 3.216 | -27 | -7 | 20 | 102 |

[a] Family-wise error corrected P < 0.0062

Note that all MNI coordinates of maximum t values are selected in the significant region.

Abbreviations: PBD, Psychotic Bipolar Disorder; NPBD, Non-Psychotic Bipolar Disorder; BD, Bipolar Disorder; HC, Healthy Control; MNI, Montreal Neurological Institute; L, Left; R, Right.

## 4.3. Local disruptions of WM connectivity along with widespread microstructural alterations in NPBD

We observed that, compared with HC, NPBD showed local decreases in FA in a part of the corpus callosum and some WM areas within the left cerebral hemisphere, including the anterior and posterior corona radiata and cingulum. However, the increase in MD was more widely distributed across both cerebral hemispheres, including the right superior corona radiata, right retrolenticular part of internal capsule, left superior longitudinal fasciculus, and left cingulum, which partially overlapped with the affected WM areas in PBD. FA measures tract directional coherence, and a decrease in FA indicates impaired WM connectivity [34]. However, MD is directionally averaged and consequently less influenced by directional coherence [35]. Thus, our findings suggest that NPBD might involve widespread WM microstructural

alterations, but WM connectivity might be compromised only in a part of the corpus callosum body and limited WM areas within the left cerebral hemisphere. In addition, major depression has been reportedly associated with microstructural alterations in the affected WM areas in NPBD in this study, particularly, the corpus callosum and corona radiata [36, 37]. There is also evidence of brain asymmetry, such as left hemispheric hypo-activation or right hemispheric hyper-activation, in patients with depression or mania [38–40]. In this respect, impairments of inter- and intra-hemispheric connectivity within the left cerebral hemisphere could be interpreted as a possible neural mechanism underlying the affective symptoms in patients with BD. Further studies to investigate hemispheric differences in the pathophysiology of BD are warranted.

### 4.4. Widespread compromise of WM integrity in BD

In this study, comparisons of HC with total BD patients, including both PBD and NPBD, showed widespread alterations of DTI parameters in major WM tracts. This is largely consistent with the results of previous DTI studies reporting extensive impairment of WM integrity across the cortico-cortical, cortico-subcortical, and callosal networks in BD [13, 14]. Thus, these findings implicate BD as a brain network disorder involving widespread compromise of WM integrity. However, it has not been elucidated whether the disruptions of WM integrity are common to both PBD and NPBD or whether they are more pronounced in PBD. Given that most patients with BD experience psychotic features in their lifetime [9], previous BD studies might have included a high proportion of patients with PBD. Thus, we supposed that the extensive WM integrity impairment reported in the previous studies might have been associated with psychotic features in patients with BD as well as their affective disturbance. In this study, comparisons between patients with PBD and HC revealed FA decrease, widely distributed across both cerebral hemispheres and a large portion of the corpus callosum. In contrast, in NPBD, this decrease was limited only to local regions within the left cerebral hemisphere and corpus callosum body. Considering that a decrease in FA reflects decreased axonal connectivity [25, 34], these findings suggest that PBD might involve extensive disruptions of inter- and intra-hemispheric WM connectivity, contributing to its psychotic features and poor clinical outcomes. However, in contrast to FA, PBD and NPBD shared increases in MD across both cerebral hemispheres, reflecting widespread alterations in axonal water diffusion in both BD subtypes. Therefore, the patterns of WM microstructural alterations in PBD and NPBD in this study should be interpreted carefully. In addition, future studies need to investigate the neurobiological substrates underlying BD by dividing the patients according to the presence or absence of psychotic features.

### 4.5. Limitations

This study has some methodological limitations. First, study participants had a wide range of illness duration and had received various types and doses of antipsychotics, mood stabilizers, or antidepressants. Considering the evidence on the effects of aging, disease severity, and medications on WM integrity [41], we cannot exclude the possibility that subject heterogeneity might have influenced the results. Second, the sample size for the PBD and NPBD groups was small and was not matched; therefore, the statistical power may not be sufficient to detect differences in the direct comparison of the two groups. Third, we could not find any significant association between illness severity and diffusion measures, probably because most patients were clinically stable with less than a mild level of mood symptoms and functional impairment.

### 4.6 Conclusion

In conclusion, widespread alterations in the WM microstructure might be a common neuro-anatomical characteristic of BD, regardless of PBD and NPBD. In particular, PBD might involve extensive disruptions of inter- and intra-hemispheric WM connectivity, suggesting that psychotic features in patients with BD might be attributed to severe impairment in brain networks. Furthermore, these findings could improve our understanding of the pathophysiological basis underlying the clinical and neurobiological continuum of psychosis. Future studies to investigate the underlying neural mechanism of psychotic features in patients with BD are warranted.

## Author Contributions

**Conceptualization:** Dong-Kyun Lee, Hyeongrae Lee, Seunghyong Ryu.

**Data curation:** Seunghyong Ryu.

**Methodology:** Dong-Kyun Lee, Hyeongrae Lee.

**Project administration:** Seunghyong Ryu.

**Supervision:** Vin Ryu.

**Visualization:** Dong-Kyun Lee, Hyeongrae Lee, Sung-Wan Kim.

**Writing – original draft:** Dong-Kyun Lee, Seunghyong Ryu.

**Writing – review & editing:** Vin Ryu, Sung-Wan Kim.

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
