## [Decision Letter · Decision Letter 0]

11 Nov 2021

PONE-D-21-31335Different patterns of white matter microstructural alterations between psychotic and non-psychotic bipolar disorderPLOS ONE

Dear Dr. Ryu,

Thank you for submitting your manuscript to PLOS ONE. After careful consideration, we feel that it has merit but does not fully meet PLOS ONE’s publication criteria as it currently stands. Therefore, we invite you to submit a revised version of the manuscript that addresses the points raised during the review process.

We look forward to receiving your revised manuscript.

Kind regards,

Lubin Wang, Ph.D.

Academic Editor

PLOS ONE

Journal Requirements:

[This research was supported by an intramural research grant from the Republic of Korea’s National Center for Mental Health (R2020-B).]

 [The funders had no role in study design, data collection and analysis, decision to publish, or preparation of the manuscript.]

Reviewers' comments:

Reviewer's Responses to Questions

**Comments to the Author**

1. Is the manuscript technically sound, and do the data support the conclusions?

Reviewer #1: Yes

Reviewer #2: No

2. Has the statistical analysis been performed appropriately and rigorously? 

Reviewer #1: No

Reviewer #2: No

3. Have the authors made all data underlying the findings in their manuscript fully available?

Reviewer #1: No

Reviewer #2: Yes

4. Is the manuscript presented in an intelligible fashion and written in standard English?

Reviewer #1: Yes

Reviewer #2: Yes

5. Review Comments to the Author

Reviewer #1: Interesting study.

Comments, please:

1- present first the analyses by comparing BD vs HC before separating them for psychosis;

2- correct the multiple comparisons for post-hoc test (Bonferroni, etc);

3- implement the analyses including ONLY right-handed BD type I patients (therefore excluding BD type II and left-handers);

4-the impact of sociodemographic (age, gender, handedness) and clinical variables (duration of illness, CPZ-equivalents, BPRS) on DTI measures has to be performed and shown - -correlations or regression should be considered;

5- BPRS can be shown in sub-clusters/sub-scales, including psychotic symptoms as well as depression-anxiety, and mania; please report them in the related table;

6- I see there are some BD type II in the PBD patients, this is strange since BD type II in general should not have any history of psychosis, how come?;

7- in general I suggest to exclude BD type II from the study and all the analyses

Reviewer #2: In this manuscript, Lee et al. conducted a DWI study that compared FA and MD measures in individuals with bipolar disorder with psychotic features, those without psychotic features and healthy controls (HC). They concluded that both patient populations significantly differed from HC in both measures, but disruption of white matter (WM) integrity was greater in PBD.

While the question of understanding the differences between the psychotic and non-psychotic BD is critically important, I am not sure that the study adequately addressed this question. First, the acquisition parameters were sub-optimal with single-shell acquisition and b=600. I am surprised that the authors were able to get some significant differences given this low b-value.

Second, considering that there were 6 contrasts of interest, the p-values had to be corrected for 6 contrasts (0.05/6=0.00833), so p<0.01 is not conservative (as the authors stated), but rather not sufficient. This correction had to be applied after the data were corrected for multiple comparisons using TFCE (or other methods). Unfortunately, the authors do not specify whether that was the case.

Third, the group comparison based on the % of voxels showing the differences with HC does not have any statistical basis and cannot serve as evidence for "disruptions of intra- and inter- hemispheric WM connectivity, in terms of FA, seem more pronounced in PBD compared to NPBD". From this perspective, the conclusions are highly overstated.

One strategy to analyze these data could 1) to compare all BD vs. HC to identify the regions affected by the disorder, and 2) use the identified regions as a mask to further compare PBD vs. NPBD.

The inclusion of BD-II to the data analyses is of concern due to a greater number of those participants in NPBD group. BD-II is even more versatile disorder than BD-I and can dilute the effects. Two resent reports on white matter in BD-II support this view by reporting different results despite using similar populations (Manelis et a., 2021 DOI: 10.1038/s41598-021-87069-2; Mak et al., 2021 DOI: 10.1038/s41598-021-81355-9). I suggest re-running the analyses using only BD-I patients.

Minor concerns include:

- please confirm that participants who experienced psychotic symptoms due to substance use were excluded

- please add the information about the data quality assurance and how many participants were removed due to poor data quality, etc.

- please add the information about whether the acquisition was multi-band and in what direction the data were collected

- please add citations for all FSL tools you used in the study

- please correct the p-values as suggested above

6. PLOS authors have the option to publish the peer review history of their article (what does this mean?). If published, this will include your full peer review and any attached files.

Reviewer #1: **Yes: **Paolo Brambilla

Reviewer #2: No

---

## [Author Response · Author response to Decision Letter 0]

15 Dec 2021

Response to the Reviewers' comments:

Reviewer #1: Interesting study.

- Response: We would like to thank the reviewer #1 for their careful and thoughtful suggestions.

Comments, please:

1- present first the analyses by comparing BD vs HC before separating them for psychosis;

- Response: Following the reviewer’s recommendation, we compared HCs and BD patients, combining PBD and NPBD. We have described the new findings and rewritten our discussion in sections 3.2 and 4.4 (highlighted, page 11 line 146–149, page 13 line 164‒167, page 18 line 261–275) as well as tables and figures.

2- correct the multiple comparisons for post-hoc test (Bonferroni, etc);

- Response: We have described in more detail the Bonferroni correction in the FWE-corrected P-value to adjust for the number of multiple comparisons in section 2.4 (highlighted, page 7 line 103–108).

3- implement the analyses including ONLY right-handed BD type I patients (therefore excluding BD type II and left-handers);

- Response: Following the reviewer’s recommendation, we re-analyzed only patients with BD type I, including 29 patients PBD and 23 NPBD. However, we had to include right-handed subjects in the study due to a significant loss of sample size when excluding right-handed ones. Instead, we statistically adjusted for handedness as a covariate to compare DTI parameters between groups. We have added more clear descriptions on the study population in sections 2.1 and 2.4 (highlighted, page 5 line 55–60, page 7 line 97–100).

4-the impact of sociodemographic (age, gender, handedness) and clinical variables (duration of illness, CPZ-equivalents, BPRS) on DTI measures has to be performed and shown - -correlations or regression should be considered;

- Response: We adjusted for age, sex, education, and handedness as covariates to compare DTI parameters between group pairs (PBD vs. HC, NPBD vs. HC, PBD + NPBD vs. HC, and PBD vs. NPBD). However, we did not adjust for clinical variables in group comparisons because clinical variables were not available in HC (highlighted, page 7 line 97–100). 

As shown in section 3.1, there were no significant differences in age, sex, level of education, and handedness between patient groups and HC or in the duration of illness, medication, symptom severity, and social functioning between the PBD and NPBD groups (highlighted, page 8 line 121–125).

In addition to group comparisons, we examined the association of DTI parameters with clinical variables including BPRS-18 and WHODAS 2.0 total scores in each PBD and NPBD group, adjusting for the chlorpromazine equivalent dose of antipsychotic drug as well as other demographic characteristics. However, we did not find any significant association with clinical variables (highlighted, page 7 line 109–114, page 14 line 175–177, page 19 line 284–287).

Nevertheless, we agree that this study could not exclude the possibility that the demographic and clinical characteristics might have influenced the results and have mentioned this point in the Limitations section (highlighted, page 19 line 280–282).

5- BPRS can be shown in sub-clusters/sub-scales, including psychotic symptoms as well as depression-anxiety, and mania; please report them in the related table;

- Response: Following the reviewer’s recommendation, we have presented the BPRS-18 subscale scores; the ‘affect,’ ‘positive symptoms,’ ‘negative symptoms,’ ‘resistance,’ and ‘activation’ proposed by Shafer (2005); as well as the WHODAS 2.0 domain scores in Table 1.

6- I see there are some BD type II in the PBD patients, this is strange since BD type II in general should not have any history of psychosis, how come?;

- Response: Using the Mini-International Neuropsychiatric Interview (MINI), we determined that eight patients with BD type II had experienced frank psychotic symptoms mainly during their depressive episode and classified them into the PBD group. According to previous studies of psychopathology on patients with BD, a significant proportion (approximately 20–40%) of patients with BP type II have experienced psychotic symptoms such as hallucination and delusion during their illness. However, following the reviewer’s recommendation, to maintain subject homogeneity, we re-analyzed only patients with BD type I (highlighted, page 5 line 55–60).

7- in general I suggest to exclude BD type II from the study and all the analyses

- Response: Following the reviewer’s recommendation, we re-analyzed only patients with BD type I, including 29 PBD and 23 NPBD (highlighted, page 5 line 55–60). Accordingly, we have rewritten our results and discussion in the Results and Discussion section. 

Reviewer #2: In this manuscript, Lee et al. conducted a DWI study that compared FA and MD measures in individuals with bipolar disorder with psychotic features, those without psychotic features and healthy controls (HC). They concluded that both patient populations significantly differed from HC in both measures, but disruption of white matter (WM) integrity was greater in PBD.

- Response: We would like to thank reviewer #2 for their careful and thoughtful suggestions.

While the question of understanding the differences between the psychotic and non-psychotic BD is critically important, I am not sure that the study adequately addressed this question. First, the acquisition parameters were sub-optimal with single-shell acquisition and b=600. I am surprised that the authors were able to get some significant differences given this low b-value.

- Response: We realized that the first submitted manuscript had some errors in the description on DTI acquisition parameters. In fact, we employed a b-value of 1000 s/mm2 in this study, not a value of 600 s/mm2. We have corrected this error in section 2.2 ‘MRI data acquisition’ (highlighted, page 5 line 68–71). 

Second, considering that there were 6 contrasts of interest, the p-values had to be corrected for 6 contrasts (0.05/6=0.00833), so p<0.01 is not conservative (as the authors stated), but rather not sufficient. This correction had to be applied after the data were corrected for multiple comparisons using TFCE (or other methods). Unfortunately, the authors do not specify whether that was the case.

- Response: We have described in more detail the FWE correction for the voxel-wise multiple testing and the additional Bonferroni correction in the FWE-corrected P-value after adjusting the number of comparisons of two DTI parameters (FA and MD) between four group pairs (PBD vs. HC, NPBD vs. HC, PBD + NPBD vs. HC, and PBD vs. NPBD) in section 2.4 ‘Statistical analyses’ (highlighted, page 7 line 103–108). 

Third, the group comparison based on the % of voxels showing the differences with HC does not have any statistical basis and cannot serve as evidence for "disruptions of intra- and inter- hemispheric WM connectivity, in terms of FA, seem more pronounced in PBD compared to NPBD". From this perspective, the conclusions are highly overstated.

- Response: We agree with the reviewer’s comment. In comparison with HC, the affected WM areas appear more widespread in PBD than in NPBD, but we could not determine the statistical differences in the area. Moreover, direct comparisons of DTI parameters between PBD and NPBD showed no significant differences. We have added descriptions on this point to section 4.3 and 4.5 (highlighted, page 17 line 253–page 19 line 256, page 19 line 282–284). In addition, we have re-written our conclusion to avoid overstating our findings (highlighted, page 15 line 191–192, page 19 line 291–294). 

One strategy to analyze these data could 1) to compare all BD vs. HC to identify the regions affected by the disorder, and 2) use the identified regions as a mask to further compare PBD vs. NPBD.

- Response: Following the reviewer’s recommendation, we compared HCs and BD patients, including PBD and NPBD. We have described the new findings and rewritten our discussion in sections 3.2 and 4.4 (highlighted, page 11 line 146–149, page 13 line 164‒167, page 18 line 261–275) as well as tables and figures.

The inclusion of BD-II to the data analyses is of concern due to a greater number of those participants in NPBD group. BD-II is even more versatile disorder than BD-I and can dilute the effects. Two resent reports on white matter in BD-II support this view by reporting different results despite using similar populations (Manelis et a., 2021 DOI: 10.1038/s41598-021-87069-2; Mak et al., 2021 DOI: 10.1038/s41598-021-81355-9). I suggest re-running the analyses using only BD-I patients.

- Response: Accordingly, we re-analyzed only patients with BD type I, including 29 PBD and 23 NPBD. We have described the study population more clearly in section 2.1 ‘Subjects’ (highlighted, page 5 line 55–60) and have rewritten our results and discussion in both sections.

Minor concerns include:

- please confirm that participants who experienced psychotic symptoms due to substance use were excluded

- Response: We confirmed that, according to the MINI, no patients had taken any kind of substance among the study subjects. We have added descriptions on this point to section 2.1 ‘Subjects’ (highlighted, page 5 line 58–59).

- please add the information about the data quality assurance and how many participants were removed due to poor data quality, etc.

- Response: We have added descriptions on this point to section 2.1 ‘Subjects’ (highlighted, page 5 line 55–56).

- please add the information about whether the acquisition was multi-band and in what direction the data were collected

- Response: In this study, we acquired diffusion-weighted images using single-shot EPI sequence. We have presented the imaging acquisition parameters in section 2.2 (highlighted, page 5 line 67–71). 

- please add citations for all FSL tools you used in the study

- Response: We have added a citation for the FSL tool in section 2.3 (highlighted, page 6 line 75–76).

- please correct the p-values as suggested above

- Response: We have described in more detail the Bonferroni correction in the FWE-corrected P-value to adjust for the number of multiple comparisons in section 2.4 ‘Statistical analyses’ (highlighted, page 7 line 105–108).

---

## [Decision Letter · Decision Letter 1]

26 Jan 2022

PONE-D-21-31335R1Different patterns of white matter microstructural alterations between psychotic and non-psychotic bipolar disorderPLOS ONE

Dear Dr. Ryu,

Thank you for submitting your manuscript to PLOS ONE. After careful consideration, we feel that it has merit but does not fully meet PLOS ONE’s publication criteria as it currently stands. Therefore, we invite you to submit a revised version of the manuscript that addresses the points raised during the review process.

We look forward to receiving your revised manuscript.

Kind regards,

Lubin Wang, Ph.D.

Academic Editor

PLOS ONE

Reviewers' comments:

Reviewer's Responses to Questions

**Comments to the Author**

1. If the authors have adequately addressed your comments raised in a previous round of review and you feel that this manuscript is now acceptable for publication, you may indicate that here to bypass the “Comments to the Author” section, enter your conflict of interest statement in the “Confidential to Editor” section, and submit your "Accept" recommendation.

Reviewer #1: All comments have been addressed

Reviewer #2: (No Response)

2. Is the manuscript technically sound, and do the data support the conclusions?

Reviewer #1: Yes

Reviewer #2: Partly

3. Has the statistical analysis been performed appropriately and rigorously? 

Reviewer #1: Yes

Reviewer #2: Yes

4. Have the authors made all data underlying the findings in their manuscript fully available?

Reviewer #1: Yes

Reviewer #2: No

5. Is the manuscript presented in an intelligible fashion and written in standard English?

Reviewer #1: Yes

Reviewer #2: Yes

6. Review Comments to the Author

Reviewer #1: amended in accordance to reviewer's comments. the manuscript can be accepted as it is in my opinion

Reviewer #2: The authors addressed most of my concerned. However, they still present the qualitative differences between the PBD vs. NPBD as statistically significant. On p.10 of the revised manuscript, the authors clearly state that "Direct comparisons of FA and MD between PBD and NPBD revealed no significant difference". Therefore, discussing "different patterns of WM connectivity disruptions between PBD and NPBD" does not make any sense. Moreover, the Discussion section should state and discuss the fact that there were NO significant differences between psychotic and non-psychotic groups. Considering that the BD groups did not differ from each other, the authors could focus on the BD vs. HC comparison. The results are also interesting and can be discussed in the context of existing literature. These results should be reported and discussed right after the non-significant results are reported and discussed. The findings regarding the HC vs. PBD and HC vs. NPBD differences can be reported, but should not be compared as the differences are not statistically significant. The statements like "In particular, disruptions of inter- and intra- hemispheric WM connectivity, in terms of FA, might be pronounced in PBD" are not statistically supported.

7. PLOS authors have the option to publish the peer review history of their article (what does this mean?). If published, this will include your full peer review and any attached files.

Reviewer #1: **Yes: **paolo brambilla

Reviewer #2: No

---

## [Author Response · Author response to Decision Letter 1]

9 Feb 2022

- Response: We would like to thank reviewer #2 for the careful and thoughtful suggestions. We agree with the comments and have revised our abstract and manuscript more modestly (highlighted, page 2 line 13–16, page 14 line 177‒178). First, we have added the negative findings of direct comparisons between PBD and NPBD in section 4.1 (highlighted, page 15 line 187–200). We have also discussed the results of comparisons with HC in sections 4.2 and 4.3. Finally, we have mentioned the implication of WM alterations in PBD, NPBD, and BD in section 4.4 (highlighted, page 17 line 187 – page 18 line 270). In addition, we have deleted the section on “different patterns of WM connectivity disruptions between PBD and NPBD.”

---

## [Decision Letter · Decision Letter 2]

7 Mar 2022

Different patterns of white matter microstructural alterations between psychotic and non-psychotic bipolar disorder

PONE-D-21-31335R2

Dear Dr. Ryu,

We’re pleased to inform you that your manuscript has been judged scientifically suitable for publication and will be formally accepted for publication once it meets all outstanding technical requirements.

Kind regards,

Lubin Wang, Ph.D.

Academic Editor

PLOS ONE

Additional Editor Comments (optional):

Reviewers' comments:

Reviewer's Responses to Questions

**Comments to the Author**

1. If the authors have adequately addressed your comments raised in a previous round of review and you feel that this manuscript is now acceptable for publication, you may indicate that here to bypass the “Comments to the Author” section, enter your conflict of interest statement in the “Confidential to Editor” section, and submit your "Accept" recommendation.

Reviewer #2: All comments have been addressed

2. Is the manuscript technically sound, and do the data support the conclusions?

Reviewer #2: Yes

3. Has the statistical analysis been performed appropriately and rigorously? 

Reviewer #2: Yes

4. Have the authors made all data underlying the findings in their manuscript fully available?

Reviewer #2: Yes

5. Is the manuscript presented in an intelligible fashion and written in standard English?

Reviewer #2: Yes

6. Review Comments to the Author

Reviewer #2: (No Response)

7. PLOS authors have the option to publish the peer review history of their article (what does this mean?). If published, this will include your full peer review and any attached files.

Reviewer #2: No

---

## [Editor Report · Acceptance letter]

10 Mar 2022

PONE-D-21-31335R2 

Different patterns of white matter microstructural alterations between psychotic and non-psychotic bipolar disorder 

Dear Dr. Ryu:

I'm pleased to inform you that your manuscript has been deemed suitable for publication in PLOS ONE. Congratulations! Your manuscript is now with our production department. 

Kind regards, 

on behalf of

Dr. Lubin Wang 

Academic Editor

PLOS ONE